# Arsenic Trioxide Triggers Apoptosis of Metastatic Oral Squamous Cells Carcinoma with Concomitant Downregulation of GLI1 in Hedgehog Signaling

**DOI:** 10.3390/biomedicines10123293

**Published:** 2022-12-19

**Authors:** Raphael Luís Rocha Nogueira, Taís Bacelar Sacramento de Araújo, Ludmila Faro Valverde, Viviane Aline Oliveira Silva, Bruno Raphael Ribeiro Cavalcante, Erik Aranha Rossi, Kyan James Allahdadi, Mitermayer Galvão dos Reis, Thiago Almeida Pereira, Ricardo D. Coletta, Daniel Pereira Bezerra, Bruno Solano de Freitas Souza, Rosane Borges Dias, Clarissa A. Gurgel Rocha

**Affiliations:** 1Gonçalo Moniz Institute, Oswaldo Cruz Foundation (IGM-FIOCRUZ/BA), Salvador 40296-710, Bahia, Brazil; 2Department of Pathology, School of Medicine of the Federal University of Bahia, Salvador 40110-909, Bahia, Brazil; 3Department of Propedeutics, School of Dentistry of the Federal University of Bahia, Salvador 40100-150, Bahia, Brazil; 4Molecular Oncology Research Center, Barretos Cancer Hospital, Barretos 14784-400, São Paulo, Brazil; 5Center for Biotechnology and Cell Therapy, D’Or Institute for Research and Education (IDOR), São Rafael Hospital, Salvador 41253-190, Bahia, Brazil; 6Institute for Stem Cell Biology and Regenerative Medicine, Stanford University, Stanford, CA 94305, USA; 7Department of Oral Diagnosis, School of Dentistry University of Campinas, Piracicaba 13414-903, São Paulo, Brazil; 8Graduate Program in Oral Biology, School of Dentistry University of Campinas, Piracicaba 13414-903, São Paulo, Brazil

**Keywords:** oral cancer, hedgehog signaling, arsenic trioxide, GLI1, drug repurposing

## Abstract

Given the lack of advances in Oral Squamous Cell Carcinoma (OSCC) therapy in recent years, pharmacological strategies to block OSCC-related signaling pathways have gained prominence. The present study aimed to evaluate the therapeutic potential of Arsenic Trioxide (ATO) concerning its antitumoral effects and the inhibition of the Hedgehog (HH) pathway in OSCC. Initially, ATO cytotoxicity was assessed in a panel of cell lines. Cell viability, cell cycle, death patterns, and cell morphology were analyzed, as well as the effect of ATO on the expression of HH pathway components. After the cytotoxic assay, HSC3 cells were chosen for all in vitro assays. ATO increased apoptotic cell death and nuclear fragmentation in the sub-G1 cell cycle phase and promoted changes in cell morphology. In addition, the reduced expression of GLI1 indicated that ATO inhibits HH activity. The present study provides evidence of ATO as an effective cytotoxic drug for oral cancer treatment in vitro.

## 1. Introduction

Among oral cancer patients, oral squamous cell carcinoma (OSCC) comprises up to 90% of all tumors [1], causing a relevant impact on patient quality of life [2]. Patients with OSCC typically present a poor prognosis, mainly when associated with late diagnosis. Typical 5-year survival rates hover around 50%, largely due to loco-regional recurrence and metastasis [3].

Embryonic signaling pathways, specifically Hedgehog (HH) pathway, have recently received increased interest, as highlighted by recent publications [4,5]. While the HH cascade is critical to the development of nearly every mammalian organ, the disruption of this cascade is clearly evidenced in several types of cancer, including OSCC [6]. Three mammalian HH ligands, Sonic (SHH), Indian (IHH) and Desert (DHH), participate in canonical activation through the Patched receptors (e.g., PTCH1 and PTCH2) and Smoothened (SMO) a transmembrane signaling protein [4]. The signals released to the cytoplasm through phosphorylation can activate one of three Glioma-associated oncogenes- GLI1, GLI2 or GLI3 [7,8]. GLI transcription factors are implicated in the activation of genes involved in cell proliferation events, such as Cyclin D1 and MYC [8], as well as angiogenesis [9], cell survival, and epithelial-mesenchymal transition [8].

The most common approach to target HH is using SMO inhibitors [10]. Nevertheless, pharmacological GLI inhibition appears to be more effective than SMO inhibition in reducing tumor proliferation and inducing apoptosis [11]. Since GLI proteins are the final effectors of the HH pathway, GLI-targeted drugs can downregulate genes and proteins related to proliferation, survival, and angiogenesis, thereby inhibiting tumor growth and therapeutic resistance [10]. In this way, the downstream mechanisms of GLI1 activation have been related to aberrant HH signaling in several human cancer [5,7], including OSCC [12]. Accordingly, GLI1 inhibitors are promising as anti-cancer drugs due to their role in targeting multiple oncogenic signaling pathways [5]. In several experimental cellular and animal models, Arsenic Trioxide (ATO) has demonstrated effectiveness against solid tumors, including lung [13], breast [14], osteosarcoma [15], and liver [16] cancer. ATO has been tested as a GLI1 inhibitor in some tumor types, such as small cell lung carcinoma [17], but not in OSCC. Accordingly, we hypothesized that ATO could decrease cell viability, and promote cell death through the inhibition of the HH pathway in OSCC. Considering the mechanisms exerted by ATO on GLI transcription factors, the present study aimed to investigate the antitumor effects of ATO on metastatic OSCC, including those related to GLI1 gene regulation.

## 2. Materials and Methods

### 2.1. Cell Culturing and Characterization

Three OSCC cell lines, HSC3 (JCRB Cell Bank, Tokyo, Japan), CAL 27 (Centre Antoine Lacassagne, Nice, France), and SCC4 (American Type Culture Collection, Manasas, VA, USA), were used in this study. In addition, human gingival keratinocyte (HGK; University of Oulu, Oulu, Finland) [18], and two different cancer-associated fibroblasts populations (CAF 1 and 2) and Normal Oral Fibroblasts (NOF) were obtained from fragments of OSCC biopsied from patients at the UNICAMP School of Dentistry (IRB approval number 4.706.681) [19].

HGK was cultured in Keratinocyte SFM medium Gibco^®^ (Life Technologies, Carlsbad, CA, USA) supplemented with penicillin/streptomycin Gibco^®^ (Life Technologies, Carlsbad, CA, USA). For all other cell lines, DMEM (Dulbecco’s modified Eagle medium) (Life Technologies, Carlsbad, CA, USA) was supplemented with 10% FBS (Fetal bovine serum, Gibco^®^) (Life Technologies, Carlsbad, CA, USA), 0.8% hydrocortisone (Sigma-Aldrich, San Louis, MO, USA) and 1% penicillin/streptomycin Gibco^®^ (Life Technologies, Carlsbad, CA, USA). All cell cultures were incubated at 37 °C under 5% CO_2_ in a humidified atmosphere. All cell lines were tested periodically for the mycoplasma presence using a luminometer, according to the MycoAlert^TM^ PLUS Mycoplasma Detection Kit (Lonza Bioscience, Morrisville, NC, USA). Tumor and non-tumor cells used in cytotoxicity assay are summarized in Appendix A.

### 2.2. Arsenic Trioxide Assessment and Treatment Protocol

In order to assess ATO cytotoxicity, an alamarBlue Assay (Thermo Fisher Scientific, Waltham, MA, USA) was performed using serial dilutions after 72 h exposure to ATO (0.96 to 126.3 µM, Sigma-Aldrich, San Louis, MO, USA). In this case, 5-fluorouracil (0.23 to 38.4 µM, 5-FU, Sigma-Aldrich, San Louis, MO, USA) was used as a positive control. To prepare the aqueous solution (5 mg/mL) of compounds, ATO (5 mg) and 5-FU (5 mg) were dissolved in DMSO PA, 99% (DMSO) (1 mL) (Dimethyl sulphoxide, Panreac). Absorbance was measured at 570 nm (oxidized/blue) and 600 nm (reduced/pink) wavelengths using a SpectraMax 190 microplate reader (Molecular Devices, San Jose, CA, USA). The calculation was obtained using the formula ARLW (percent of reduced cells) = ALW (oxidized cells in 570 nm) − (AHW/reduced cells in 600 nm × RO) × 100, in which R0 (correction factor) = AOLW/AOHW (blank samples), according to the alamarBlue Assay (Thermo Fisher Scientific, Waltham, MA, USA). The percentage of inhibition of the tested drugs (Arsenic Trioxide and 5-FU) was calculated using the mean of cell viability of the negative group control (DMSO) as the reference group. For this assay, three independent experiments were carried out in duplicate.

### 2.3. Membrane Integrity Assessment Assay Using Trypan Blue

The membrane integrity of HSC3 cells was determined using the trypan blue exclusion method and counted in a Neubauer chamber under direct microscopy (Olympus CX41) to evaluate viable cells. All results were confirmed by at least three independent experiments performed in duplicate.

### 2.4. Cell Cycle and Internucleosomal DNA Fragmentation

Cell cycle analysis was performed by flow cytometry using PI (propidium iodide) as a fluorogenic agent. After treatment with ATO or 5-FU, cells were resuspended in permeabilization solution (0.1% triton X-100, 0.1% sodium citrate, 2 μg/mL PI and 100 µg/mL RNase in distilled water) and were analyzed on a BD LSRFortessa^®^ flow cytometer using BD FACSDiva software version 6.2 (Becton Dickinson Biosciences, San Jose, CA, USA). FlowJo software version 10 (FlowJo LCC, Ashland, OR, USA) was used to evaluate the proportions of fragmented internucleosomal DNA and determine cell cycle phases. Cellular debris was omitted from the analyses and 10,000 events were analyzed per sample. For this experiment, three independent experiments were carried out in duplicate.

### 2.5. Cell Death, Viability, and Morphology Evaluations

First, to evaluate cellular viability and death after 48 and 72 h of treatment, HSC3 cells were labeled with annexin V-FITC and Propidium Iodide (PI) in accordance with the manufacturer’s protocol (BD Biosciences, Franklin Lakes, NJ, USA). Frontal Light Scatter and Side Scatter were evaluated on a BD LSRFortessa^®^ flow cytometer using FACSDiva software version 6.2 (Becton Dickinson Biosciences, San Jose, CA, USA). FlowJo software, version 10 (FlowJo LCC, Ashland, OR, USA) was used to assess the proportion of cells undergoing apoptosis and cell morphology. Cellular debris was omitted from the analyses and 10,000 events were analyzed per sample. Three independent experiments were carried out in duplicate.

In order to evaluate cell death and viability through fluorescence staining, annexin V-FITC and Propidium Iodide (PI) (BD Biosciences, Franklin Lakes, NJ, USA) were used after 48 and 72 h of treatment. Cells were centrifuged and not washed. Next, 100 µL of binding buffer solution containing 3 µL of annexin V-FITC and 3 µL of PI was added. After a 15 min incubation in the dark, an additional 100 µL of binding buffer was added to each sample. Doxorubicin (1 µg/mL) was added to evaluate the reaction. The images were acquired on an A1+ Singlephoton Confocal Microscope (Nikon, Tokyo, Japan).

### 2.6. Western Blot

After 24 h of treatment, cells representative of each experimental group were lysed with 50 mM of Tris-HCl buffer pH 7.4 containing 1% Triton X-100, 150 mM NaCl, 0.5 mM EGTA, 0.5 mM EDTA and an anti-protease cocktail (Complete Protease Inhibitor Cocktail Tablets, Roche, France). The protein extracts (30 µg) were separated by SDS-PAGE and transferred using iBlot Gel Transfer Stacks Nitrocellulose Mini (Invitrogen, Thermo Fisher Scientific). Subsequently, membranes were blocked with 5% milk powder in TBS/0.1% Tween (20 mM Tris, 150 mM NaCl, 0.1% Tween 20, pH 7.6) and then incubated with the corresponding primary antibodies (rabbit polyclonal GLI1 (1:1000, Sigma Aldrich, San Louis, MO, USA) and ß-actin (1:10,000, Sigma Aldrich, San Louis, MO, USA) in 5% BSA in TBS/0.1% Tween overnight at 4 °C. The membranes were then washed with TBS-T and incubated with the appropriate horseradish peroxidase (HPR)–conjugated antibodies (anti-mouse IgG (1:10,000, Santa Cruz Biotech, Texasdallas, TX, USA) and anti-rabbit IgG (1:10,000, GE Health) in blocked solution at room temperature for 1 h. Signal detection was performed on an ECL + Chemiluminescence Detection System (PerkinElmer, Villebon-sur-Yvette, France) using ImageQuant^TM^ LAS 4000 system (GE Healthcare, Chicago, IL, USA). The labeled bands were analyzed and quantified using the Image J software (National Institutes of Health, Bethesda, MD, USA). The data shown are representative of three independent experiments.

### 2.7. Immunofluorescence

After 48 h of treatment, HSC3 cells were washed in sterile 1 × PBS (pH 7.2) and fixed for 10 min. in ice-cold methanol. After washing, non-specific site blocking cells were performed using 10 mL PBS 1×, 30 μL Triton X-100, 250 μL FBS and 0.3 g BSA for 30 min. Rabbit polyclonal antibody against GLI1 (1:500, Novus Biologicals, Englewood, CO, USA) was incubated overnight. The cells were labeled with Alexa Fluor 488 rabbit IgG (1:800, Thermo Fisher) and cell nuclei were stained with 5 μg/mL of 4,6-diamidino-2-phenilindole dihydrochloride (DAPI, Molecular Probes, Eugene, OR). The images were acquired on an A1+ Singlephoton Confocal Microscope (Nikon, Tokyo, Japan). The ImageJ software was used to analyze the Fluorescence Intensity to determine the mean of gray value (MGV). Next, the Fluorescence Intensity was calculated as follows: Fluorescence Intensity = MGV/Number of cells. For immunofluorescence, three independent experiments were performed.

### 2.8. Gene Expression of HH Pathway Components

In order to perform total RNA isolation, HSC3 cells were cultured and incubated overnight, followed by treatment for 24 h. All experiments were carried out under DNAse/RNAse-free conditions. RNA was extracted using a RNeasy Plus Mini Kit (QIAGEN, Hilden, Germany). The quantity and purity of the RNA extracted from each sample was analyzed on a NanoDrop™ Lite Spectrophotometer (Thermo Fisher Scientific, Waltham, MA, USA). Reverse transcription amplification was performed using Superscript VILO™ transcriptase enzyme (Invitrogen Corporation, Waltham, MA, USA). cDNA samples were stored at −20 °C.

The expression of components of the HH pathway was evaluated using qPCR via the TaqMan Gene Expression Assays™ inventoried for GLI1 (Hs01110766_m1), GLI2 (Hs01119974_m1) and GLI3 (Hs0060923_m1). After testing using an endogenous gene panel, GAPDH (Hs99999905_m1) was chosen as the reference gene. The reactions were performed using 96-well plates with a volume of 20 μL per well on the ViiA 7 Real-Time PCR System (Applied Biosystems™, Foster City, CA, USA). Each well contained 10 ng/μL of sample cDNA (8 μL), 1 μL of target gene solution (Applied Biosystems^TM^, Foster City, CA, USA), 10 μL of TaqMan™ Fast Advanced Master Mix (Applied Biosystems, Foster City, CA, USA) and 1 μL of RNAse-free water. The amplification protocol consisted of an initial step consisting of 50 °C for 2 min and 95 °C for 10 min, followed by 40 cycles at 95 °C for 15 s and 60 °C for 1 min. The quantification cycle (Cq) values were obtained using ExpressionSuite Software Version 1.3 (Applied Biosystems, Foster City, CA, USA). In order to perform relative quantification, the Cq comparative method (2^−ΔΔCq^) was used. Each sample’s Cq value was normalized using the geometric mean of the Cq values obtained using the GAPDH reference gene and calibrated using the geometric mean of the Cq values from non-treated HSC3 cells (HSC3 NT). For this assay, three independent experiments were carried out in duplicate.

### 2.9. Statistical Analysis

The results were analyzed using GraphPad Prism software version 6.03 (GraphPad Software, Inc., San Diego, CA, USA), considering a “*p*” value corresponding to alpha (α) less than or equal to 5%. IC_50_ values were obtained using non-linear regression, considering three independent experiments carried out in duplicate (similar experimental conditions). The differences between groups in vitro assays were evaluated by ANOVA (analysis of variance) testing, followed by the Student-Newman-Keuls test (*p* ≤ 0.05).

## 3. Results

### 3.1. ATO Exerts Cytotoxic Effects in a Metastatic Lineage of OSCC

The cytotoxic activity of ATO was evaluated in a panel of tumor and non-tumor cells (Appendix A). The lowest IC_50_ value was obtained using the HSC3 cell line (37.4 µM). Since HSC3 cells demonstrated greater sensitivity to ATO, with a corresponding selective index (SI) value of 3.98 in comparison to HGK, this cell line was used in all following in vitro assays. Treatment with ATO revealed IC_50_ (Half-maximal Inhibitory concentration) values ranging from 0.28 (CAF 1) to greater than 126.4 (in the CAL 27, SCC4, CAF 2, HGK and NOF lines). Cytotoxicity was also observed using the positive control, 5-FU, with resulting IC_50_ values ranging from 0.07 µM (SCC4) to over 192 µM (CAF 1 and 2) (Table 1).

### 3.2. ATO Reduces Membrane Integrity of HSC3 Cells

Treatment with ATO significantly reduced membrane integrity at all treatment times (24, 48 and 72 h) compared to the negative control and non-treated cells (Figure 1). Regarding cell viability as determined by the annexin V-FITC/propidium iodide (PI) apoptotic assay, ATO at a concentration of 37.4 µM significantly reduced the percentage of viable cells after 48 h of incubation compared to cells inoculated with the DMSO vehicle and non-treated cells. Following 72 h of incubation, a significant reduction in viability was also observed (at both 37.4 µM and 18.7 µM of ATO), demonstrating a dose-dependent effect (Appendix A).

### 3.3. ATO Promotes Cell Death by Apoptosis through Significantly Increased Nuclear Fragmentation

An increase in cell populations in the sub-G1 phase was observed at 24, 48 and 72 h of treatment with 37.4 μM of ATO (Figure 2). A significant increase in cell populations in the Sub-G1 phase was also found at 72 h with the lower concentration (18.7 μM) of ATO. In addition, the presence of cells in sub-G1 phase in the cell cycle assay could be related to the presence of internucleosomal DNA fragmentation, as a strong indication of apoptotic death is demonstrated (Appendix A). Our evaluation of cell death using Annexin V-FITC/PI indicated an increased percentage of cells in the early and late apoptotic stages after ATO treatment compared to negative control (CTL; DMSO 0.25%) and the non-treated group (Figure 3). Despite higher percentages of necrosis seen within 48 h of treatment at both concentrations of ATO (37.4 µM and 18.7 µM) compared to controls (CTL) or HSC3 NT, a significant difference was only detected at 72 h using the lower concentration (18.7 µM) of ATO (Appendix A). In addition, Figure 4 represents cell populations in early and late apoptosis and necrosis (annexin/PI immunofluorescence images) and Appendix A represents the control (Doxorubicin) used for this reaction.

### 3.4. ATO Promotes Cell Shrinkage, Increases Granularity and Nuclear Condensation in Metastatic OSCC

ATO treatment promoted cell shrinkage in HSC3 cells, as demonstrated by decreased forward scatter (FSC) as well as increased granularity and nuclear condensation, evidenced by enhancement in side scatter (SSC), especially after 72 h of treatment in the highest concentration. These effects on morphology were shown to be concentration- and time-dependent, consistent with mechanisms of cell death (Figure 5 and Appendix A).

### 3.5. Reduction of GLI Protein and mRNA Expression after ATO Treatment

Treatment with ATO not only weakened the immunoexpression of GLI1 compared to the other groups evaluated, but also promoted alterations in cell morphology (Figure 6A,B). Decreased mRNA levels of GLI1 were observed after 24 h of treatment with ATO at 37.4 µM (Figure 6C). In addition, reduction in mRNA levels were seen within GLI2 and GLI3 transcription factors compared to the negative control (DMSO 0.25%) and HSC3 NT cells (Appendix A).

In addition, the evaluation of GLI1 protein after ATO treatment at both tested concentrations revealed a reduced expression of GLI1 (Figure 7 and Appendix A), confirming our mRNA transcript expression results (Figure 6C).

## 4. Discussion

In an effort to contribute to research surrounding the repositioning of drugs that can act in the adjuvant treatment of OSCC, the antitumor effects of ATO and the pharmacological inhibition of the HH pathway through GLI were evaluated in a metastatic OSCC cell line (HSC3) using in vitro assays.

Initially, ATO cytotoxicity was evaluated in a panel of tumor and non-tumor cells, with the HSC3 (metastatic OSCC) cell line demonstrating the lowest IC_50_ of all evaluated tumor cells. Among the cell lines tested against ATO, CAF 1 was the most sensitive, while NOF offered the least sensitivity. The former result seems promising, since studies in the literature have demonstrated that cancer-associated fibroblasts represent the most abundant cell population in the tumor microenvironment [20,21], and this cell type is often associated with parameters indicative of a severely deleterious prognosis, e.g., vascular invasion, metastatic dissemination, and tumor recurrence [22].

The immunofluorescence analysis revealed that ATO treatment induces morphological changes in the HSC3 cell line. The cells appeared more spherical and smaller when treated with the higher concentration of ATO evaluated. This finding is supported by a study evaluating treatment with ATO and Cisplatin for 48 h in the HSC 2 lineage, which demonstrated similar changes in cell morphology [23]. Flow cytometry analysis of ATO-treated cells confirmed our previous results indicating changes in light scattering (FSC/SSC) evidenced alterations compatible with the induction of apoptosis, i.e., reduced cell size and increased granularity. In addition, other antiproliferative properties exerted by ATO, such as wound healing, are demonstrated in pancreatic cancer cells [24], as well as in breast cancer cells [25].

Our results indicate that the higher concentration of ATO increased the percentage of cells undergoing apoptosis at all times evaluated, which was most expressive at 72 h of treatment. In this context, similar results have been described in OSCC, in which treatment with ATO for 72 h induced apoptosis in SCC9, SCC25, CAL27, and FADU cell lines [26]. In addition, ATO increased the cancer cells’ sensitivity to death receptor-induced apoptosis [27].

In the present study, we showed activation of the HH pathway in HSC3 cells and reduction of GLI protein and genes after ATO treatment. Similarly, Linder et al. (2019) [28] demonstrated that isolated treatment with ATO was capable of reducing the expression of genes involved in the HH pathway, such as GLI1. In osteosarcoma [15], colon carcinoma [29] and rhabdomyosarcoma cells [30], ATO also inhibited the transcription of GLI genes, promoting apoptotic cell death.

We have speculated that the relationship between ATO-induced apoptosis and GLI1 downregulation could be explained by the DNA damage pathway. It has been reported that ATO treatment induces cell death and DNA damage, as described by Nakamura et al. (2013) [15] in human osteosarcoma. Therefore, Nakamura et al. (2013) [15] investigated whether HH signaling activation affects the accumulation of DNA damage. The authors demonstrated that the treatment with Sonic HH attenuated the upregulation of γH2AX and ATO treatment reverted the attenuation of DNA damage caused by HH activation. These findings suggest that ATO promotes the accumulation of DNA damage by inhibiting HH signaling. Further, considering recent findings that implicated the autophagic cell death mediated by ATO in osteosarcoma cells and knowing a possible link between HH signaling and autophagy, we cannot exclude the contribution of other pathways [31,32]. Finally, despite these findings, the mechanism (s) that underlies ATO-mediated apoptosis remains elusive and additional studies are warranted to establish this point. It is noteworthy that one important limitation of the present study is using one single cell line of the evaluated tumor type for the functional assay. Considering the importance of heterogeneity in the cancer context, it would be desirable to use more than one cell line that recapitulates this heterogeneity and complexity of cancer. Therefore, additional studies are warranted to address this topic.

## 5. Conclusions

In this study, we demonstrated that ATO presents the therapeutic potential in metastatic OSCC and that one potential mechanism of action occurs through of inhibit cell viability and altering cellular morphology, promoting increased nuclear fragmentation and apoptotic cell death. In addition, ATO was able to promote HH activity inhibition. Our investigation represents the first attempt to describe HH inhibition by ATO in OSCC. However, as the relationship between ATO-induced apoptosis and GLI1 downregulation is not yet fully understood, further studies are needed to comprehensively assess the pharmacological properties of this drug in the context of OSCC. Studies that look at understanding DNA damage or/and autophagy involvement, CAF cells using in vivo models, and those that mimic the microenvironment of OSCC, including combined therapy with other HH inhibitors (e.g., vismodegib) and drugs (e.g., cisplatin) are desirable. It is noteworthy that although the involvement of GLI1 and HH pathway in HSC-3 is also supported by our previous study using GANT-61 inhibitor [12], additional GLI1 gene editing studies could constitute the proof-of-concept of its role in the cancer context.

## Figures and Tables

**Figure 1 biomedicines-10-03293-f001:**
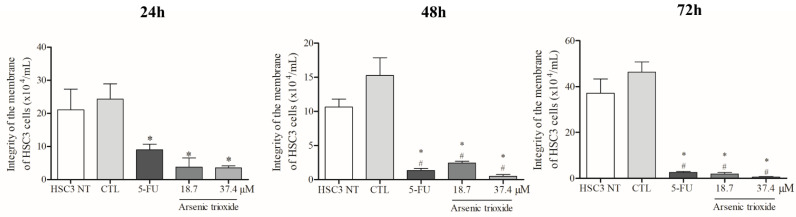
The cytotoxic effect of ATO on membrane integrity of HSC3 cells. Numbers of cells with integrity of membrane correspond to mean ± SEM of three independent experiments carried out in duplicate. (*) *p* ≤ 0.05 compared to the negative control (CTL), and (#) when compared to the HSC3 NT group (non-treated cells) by ANOVA (analysis of variance) followed by Student Newman-Keuls test.

**Figure 2 biomedicines-10-03293-f002:**
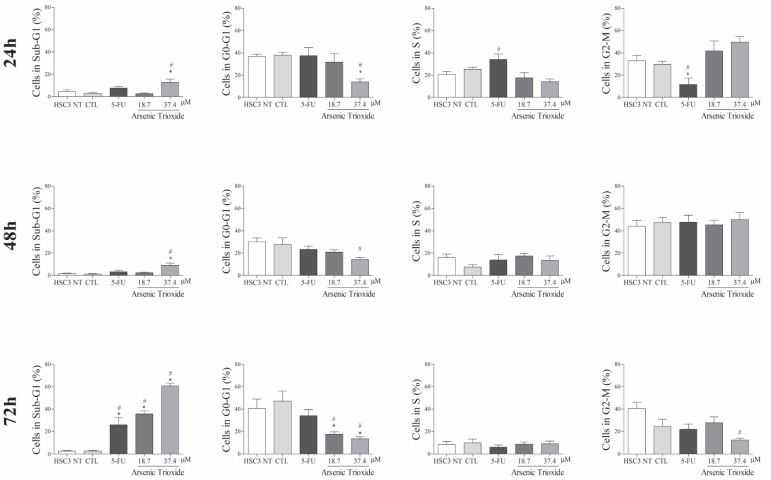
The effects of ATO on cell cycle phases (Sub-G1, G0-G1, S and G2-M) and internucleosomal DNA fragmentation in HSC3 cells at 24, 48 and 72 h of treatment. The vehicle was used (0.25% DMSO) as the negative control (CTL) and 5-FU (16.75 µM) as the positive control. The values correspond to means ± SEM from three independent experiments carried out in duplicate. Cellular debris was omitted from the analyses and 10,000 events were analyzed per sample. (*) *p* ≤ 0.05 when compared to the negative control (CTL), and (#) when compared to the HSC3 NT group (non-treated cells) by ANOVA (analysis of variance) followed by Student Newman-Keuls test.

**Figure 3 biomedicines-10-03293-f003:**
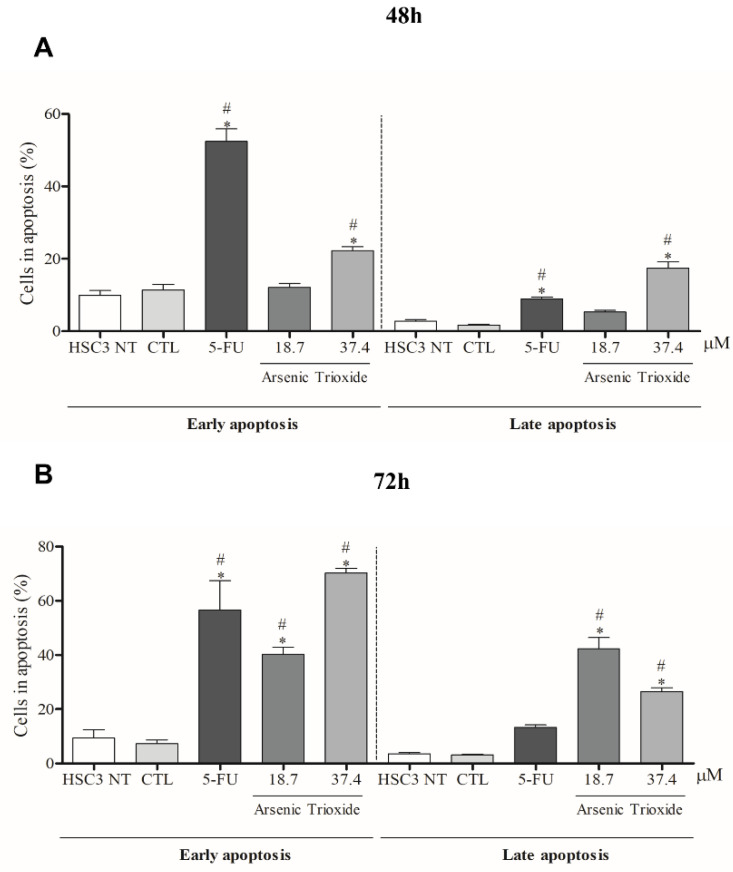
The effects of ATO on the externalization of phosphatidylserine in HSC3 cells, determined by flow cytometry using Annexin V-FITC/Propidium iodide (Early apoptosis/Late apoptosis), after 48 (**A**) and 72 h (**B**) of treatment. Cellular debris was omitted from the analyses and 10,000 events were analyzed per sample. (*) *p* ≤ 0.05 when compared to the negative control (CTL) and (#) when compared to the HSC3 NT group (non-treated cells) by ANOVA (analysis of variance) followed by Student Newman-Keuls test.

**Figure 4 biomedicines-10-03293-f004:**
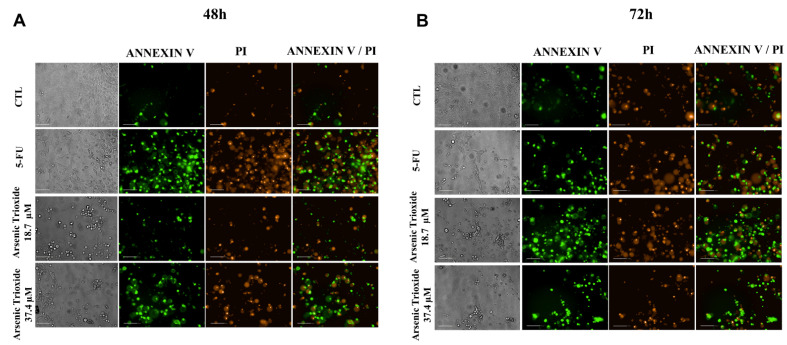
Bright field and representative images of HSC3 cells under early apoptosis, late apoptosis and necrosis after 48 h (**A**) and 72 h (**B**) of treatment, using annexin V-FITC (Green)/Propidium Iodide (PI; Red) (Scale Bar: 100 µm).

**Figure 5 biomedicines-10-03293-f005:**
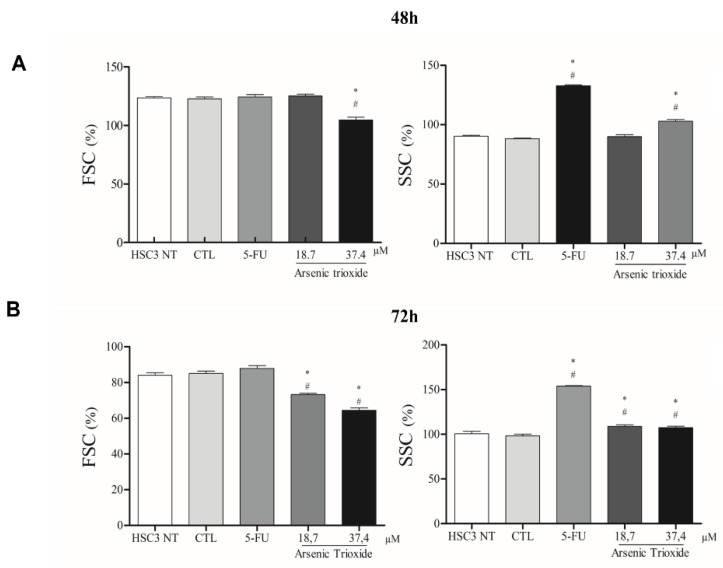
The graphical representation of light scattering characteristics of HSC3 cells, as determined by flow cytometry after 48 h (**A**) and 72 h (**B**) of treatment. The percentage (%) of FSC (Forward Scatter) and SSC (Side Scatter) correspond to mean ± SEM of three independent experiments carried out in duplicate. (*) *p* ≤ 0.05 compared to the negative control (CTL), and (#) when compared to the HSC3 NT group (non-treated cells) by ANOVA (analysis of variance) followed by Student Newman-Keuls test.

**Figure 6 biomedicines-10-03293-f006:**
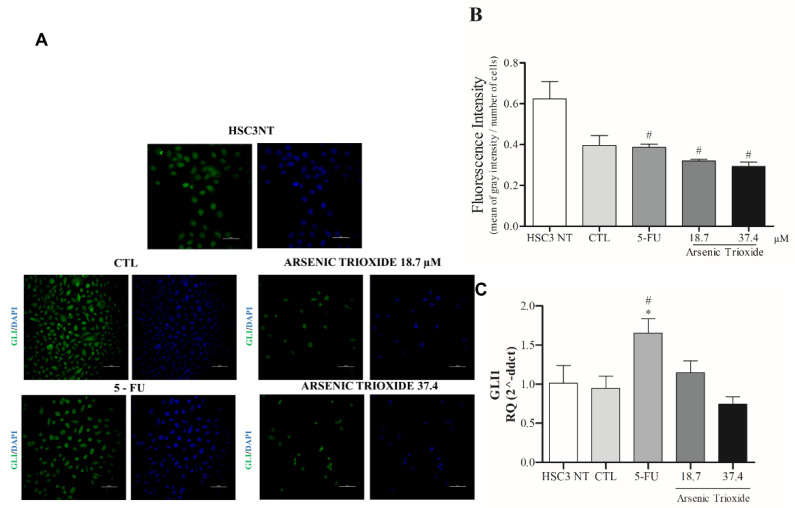
The representative images of GLI1 protein staining (green) in HSC3 cells. Nuclei are in blue. (Scale Bar: 100 µm) (**A**). Graphic representation of Fluorescence Intensity (**B**). Gene expression of GLI1 (**C**). The vehicle was used (0.25% DMSO) as a negative control (CTL) and 5-FU (16.75 μM) was used as a positive control. The Cq value for each sample was normalized using GAPDH as a reference gene, then calibrated according to the Cq values obtained for the untreated HSC3 cell group (HSC3 NT). (*) *p* ≤ 0.05 when compared to the negative control (CTL), and (#) when compared to the HSC3 NT group (non-treated cells) by ANOVA (analysis of variance) followed by Student Newman-Keuls test.

**Figure 7 biomedicines-10-03293-f007:**
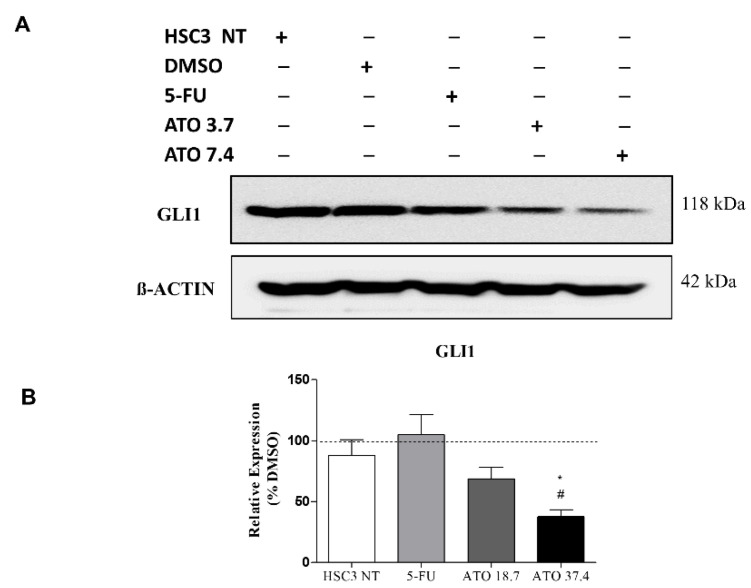
The evaluation of GLI1 protein expression in HSC3 cells after 24 h of treatment with ATO. (**A**) Low GLI1 protein expression levels in HSC3 cells treated with ATO. The symbol (−) represents the absence and (+) represents the experimental group in each band in the following order: HSC3 NT (non-treated), DMSO, 5-FU, ATO 3.7 µM and ATO 7.4 µM. (**B**) Graphic representation of densitometry readings of three individual experiments. The value of each experimental condition was calibrated using GLI1/ß-actin intensity ratio and then DMSO (---) was used as the reference group. (*) *p* ≤ 0.05 when compared to HSC3 NT (non-treated cells) group and (#) when compared to 5-FU by ANOVA (analysis of variance) followed by Student Newman-Keuls test.

**Table 1 biomedicines-10-03293-t001:** The cytotoxic activity of Arsenic Trioxide and 5-FU in tumor and non-tumor cells. The data presented as IC_50_ values corresponding to μM (micromolar) by non-linear regression analysis of three independent experiments performed in duplicate (Alamar Blue assay, 72 h of incubation).

	Arsenic Trioxide	5-FU
Tumor cells		
CAL 27	>126.4	30.8
HSC 3	37.4	16.75
SCC 4	>126.4	0.07
Cancer-associated fibroblasts		
CAF 1	0.28	>192
CAF 2	>126.4	>192
Non-tumor cells		
HGK	>126.4	>192
NOF	>126.4	39.59

## Data Availability

All data created or analyzed during this study are available from the corresponding author on reasonable request.

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
