# Peer review of "Arsenic Trioxide Triggers Apoptosis of Metastatic Oral Squamous Cells Carcinoma with Concomitant Downregulation of GLI1 in Hedgehog Signaling"

_biomedicines, 2022, doi:10.3390/biomedicines10123293_

Round 1

Reviewer 1 Report

The authors investigated the effect of arsenic trioxide (ATO) on the inhibition of the Hedgehog (HH) pathway in oral squamous cell carcinoma (OSCC). They showed that ATO induced apoptosis and increased the cell population in the sub-G1 cell cycle phase and inhibited the cell migration. In addition, ATO decreased the protein expression of GLI1. Collectively, the author concluded that ATO might be effective cytotoxic drug for oral cancer treatment in vitro.  The author must examine the additional experiments to make the author's interpretations justified.

Major comments

1. The authors only utilized single OSCC cell line HSC-3 throughout in the biological experiment except for cytotoxicity. The author must use one more cell line for the evaluation regarding the effect of ATO on HH pathway in OSCC cell lines.

2. In Materials and Methods section, the authors described the concentration unit of ATO as follows: “ATO (0.19 to 25 μg/mL, Sigma-Aldrich)” However, the author showed the IC50 as μM. The authors should explain why the concentration unit of ATO in the Methods section is not consistent with that in the other sections.

3. Related to the comment2, the author should describe how to prepare the aqueous solution of ATO in the Materials and Methods section.

4. In migration assay, the doses of ATO were set at 37.4 and 74.8 μM, and such doses of ATO was shown to readily induce apoptosis in HSC-3 cells. Therefore, the author should evaluate the migration activity in lower doses than those utilized in the apoptosis assay. Otherwise, it would be quite difficult to state ATO suppresses migration activity in vitro.

5. Does knockdown of GLI1 cause apoptosis, cell cycle arrest, or suppression of migration? The reviewer doubts the expression of GLI1 is really important in the HSC-3 cell line tested. The author should evaluate the involvement of GLI1 in the cell survival and/or cell mobility in vitro.

Author Response

Please see the attachment: Letter of resubmission + Response to Reviewer 1.

Reviewer 2 Report

The paper by Rocha Nogueira analyses the effects of Arsenic Trioxide on metastatic oral squamous carcinoma cells. 

The introduction is catchy and well-written, but in my opinion, it seems like there's a lack of background on ATO and on what led the authors to choose this molecule. 

Materials and methods are clearly described.

Discussions and conclusions seem to answer all the readers' questions and suggest the aim for further studies.

Author Response

Please see the attachment: Letter of resubmission + Response to Reviewer 2.

Round 2

Reviewer 1 Report

Thank you for the prompt revision. The authors should only examine the effect of GLI inhibitor, such as GANT 61, on the apoptosis of HSC-3 cells, instead of using knockdown experiment. The author should show the relationship between ATO-induced apoptosis and GLI1 downregulation.

Round 3

Reviewer 1 Report

The reviewer suggests changing the title of the study (e.g. "ATO triggers apoptosis of metastatic oral squamous cell carcinoma with concomitant downregulation of GLI1 in hedgehog signaling). Because the relationship between ATO-induced apoptosis and downregulation of GLI1 remains obscure. 
